# Exploration of the Gut Microbiome in Thai Patients with Major Depressive Disorder Shows a Specific Bacterial Profile with Depletion of the *Ruminococcus* Genus as a Putative Biomarker

**DOI:** 10.3390/cells12091240

**Published:** 2023-04-25

**Authors:** Michael Maes, Asara Vasupanrajit, Ketsupar Jirakran, Pavit Klomkliew, Prangwalai Chanchaem, Chavit Tunvirachaisakul, Sunchai Payungporn

**Affiliations:** 1Department of Psychiatry, Faculty of Medicine, Chulalongkorn University and King Chulalongkorn Memorial Hospital, The Thai Red Cross Society, Bangkok 10330, Thailand; 2Department of Psychiatry, Medical University of Plovdiv, 4002 Plovdiv, Bulgaria; 3Research Institute, Medical University Plovdiv, 4002 Plovdiv, Bulgaria; 4Kyung Hee University, 26 Kyungheedae-ro, Dongdaemun-gu, Seoul 02447, Republic of Korea; 5Maximizing Thai Children’s Developmental Potential Research Unit, Department of Pediatrics, Faculty of Medicine, Chulalongkorn University, Bangkok 10330, Thailand; 6Center of Excellence in Systems Microbiology, Department of Biochemistry, Faculty of Medicine, Chulalongkorn University, Bangkok 10330, Thailand

**Keywords:** major depression, bacterial translocation, gut–brain axis, neuro-immune, inflammation, oxidative and nitrosative stress, microbiome

## Abstract

Maes et al. (2008) published the first paper demonstrating that major depressive disorder (MDD) is accompanied by abnormalities in the microbiota–gut–brain axis, as evidenced by elevated serum IgM/IgA to lipopolysaccharides (LPS) of Gram-negative bacteria, such as *Morganella morganii* and *Klebsiella Pneumoniae*. The latter aberrations, which point to increased gut permeability (leaky gut), are linked to activated neuro-immune and oxidative pathways in MDD. To delineate the profile and composition of the gut microbiome in Thai patients with MDD, we examined fecal samples of 32 MDD patients and 37 controls using 16S rDNA sequencing, analyzed α- (Chao1 and Shannon indices) and β-diversity (Bray–Curtis dissimilarity), and conducted linear discriminant analysis (LDA) effect size (LEfSe) analysis. Neither α- nor β-diversity differed significantly between MDD and controls. *Rhodospirillaceae*, *Hungatella*, *Clostridium bolteae*, *Hungatella hathewayi*, and *Clostridium propionicum* were significantly enriched in MDD, while Gracillibacteraceae family, *Lutispora*, and *Ruminococcus genus*, *Ruminococcus callidus*, *Desulfovibrio piger*, *Coprococcus comes*, and *Gemmiger* were enriched in controls. Contradictory results have been reported for all these taxa, with the exception of *Ruminococcus*, which is depleted in six different MDD studies (one study showed increased abundance), many medical disorders that show comorbidities with MDD, and animal MDD models. Our results may suggest a specific profile of compositional gut dysbiosis in Thai MDD patients, with increases in some pathobionts and depletion of some beneficial microbiota. The results suggest that depletion of *Ruminococcus* may be a more universal biomarker of MDD that may contribute to increased enteral LPS load, LPS translocation, and gut–brain axis abnormalities.

## 1. Introduction

There is now evidence that major depressive disorder (MDD) is a neuro-immune, neuro-oxidative, and neuro-nitrosative (NINONS) disorder characterized by: (a) activation of the immune-inflammatory response system (IRS) and a relative deficit in the compensatory immune-regulatory system (CIRS), and T-regulatory (Treg) cell functions, which tend to attenuate an overzealous IRS; (b) activation of oxidative pathways causing damage to lipids, proteins and DNA, and IgM/IgG-mediated autoimmune responses to self-antigens and oxidative specific epitopes; (c) hypernitrosylation with increased IgM-mediated autoimmune responses to nitrosylated proteins; and (d) lowered antioxidant levels, including lowered high-density lipoprotein and lowered lecithin acyl transferase and paraoxonase-1 (PON1) activity [1,2,3,4,5,6,7,8].

Despite the fact that MDD is a NINONS-associated disorder, the primary question is what causes these pathophysiological deviations [9]. Major contributors are psychosocial stressors, particularly adverse childhood experiences [8], genetic polymorphism, for example, the PON1 Q192 gene variant [8], nutritional factors, including lowered omega-3 polyunsaturated fatty acids [4], zinc, and vitamin D [10], tobacco use disorder [9], metabolic aberrations [11], and increased load of lipopolysaccharides (LPS) due to periodontitis [12] or translocation of Gram-negative gut-commensal bacteria via increased permeability of the intestinal barrier [13].

In fact, the first paper indicating that MDD is associated with alterations in the gut microbiota–brain axis was published in 2008, stating that many, but not all, MDD patients exhibit increased translocation of Gram-negative enterobacteria or their LPS as assessed with increased serum levels of IgM/IgA to the LPS of *Morganella morganii*, *Hafnia alvei*, *Citrobacter Koseri*, *Pseudomonas Putida*, *Pseudomonas Aeruginosa*, and *Klebsiella Pneumoniae* [13]. In addition, we discovered that in numerous case reports, these elevated IgM/IgA responses to LPS were accompanied by indicators of gut dysbiosis, such as a dysbalance in the gut flora and changes in secretory IgA, β-defensin, α-antitrypsin, and calprotectin levels in stool [14,15]. The increased bacterial translocation in MDD is associated with greater severity of depression and irritable bowel symptoms, and is frequently associated with small intestine bacterial overgrowth (SIBO), as well as food, lactose, fructose, and gluten intolerances, according to case reports [14,15]. Importantly, in MDD, there are highly significant associations between IgA/IgM responses to LPS of gut-commensal bacteria (indicating leaky gut) and NINONS and autoimmune pathways, indicating that bacterial translocation in MDD may drive, at least in part, the activated NINONS pathways [16]. The stimulation of the Toll-Like Receptor-4 (TLR4) complex by LPS, resulting in the activation of NINONS pathways, is one mechanism by which leaky gut may produce depressive behaviors [17]. Moreover, TLR4 gene polymorphisms are associated with MDD [18] and increased stress-induced bacterial translocation stimulates CNS neuro-inflammatory pathways in a rodent depression model [19].

Recent studies in MDD have supported the primary findings of Maes et al. [13,14,16]. For instance, in MDD, greater intestinal permeability as measured by the lactulose/mannitol test is substantially related to depression severity [20]. MDD is also characterized by other biomarkers of leaky gut, such as elevated LPS-binding protein and intestinal fatty acid-binding protein levels in association with inflammatory biomarkers and higher depression severity [21,22]. A study published in Science, of how diet modulates the microbiome, observed increased levels of *Morganella* and *Klebsiella* in association with mental depression [23]. MDD is related to leaky gut indicators in patients with inflammatory bowel disease [24], and, in MDD, bacterial translocation is accompanied by a reduction in the CIRS functions, namely, the number of Treg cells, thereby contributing to inflammatory responses [25]. A recent meta-analysis shows that major neuropsychiatric disorders, including a depressive episode, schizophrenia, and chronic fatigue syndrome, are characterized by increased zonulin (four studies), LPS (two studies), antibodies to LPS (seven studies), sCD14 (six studies), and LPS-binding protein (two studies) [25].

Several etiologic factors can lead to leaky gut, including inflammatory processes, oxidative stress, nutritional factors, use of antibiotics and nonsteroid anti-inflammatory drugs, alcohol abuse, chemotherapy, surgery, radiation, viral infections including HIV, IBD, autoimmune disorders, and compositional gut dysbiosis with variations in gut microbiota communities [26]. Changes in gut microbial populations are now proposed to be strongly implicated in the pathophysiology of MDD [27,28]. Using second-generation sequencing of bacterial 16S RNA genes, for instance, it was discovered that the alpha diversity of gut bacteria was lower in MDD than in controls [29]. A recent review demonstrates that MDD is associated with a disparate representation of bacterial genus in comparison to controls, including increases in *Klebsiella*, *Clostridium*, *Blautia*, *Parabacteroides*, *Parasutterella*, *Streptococcus*, *Anaerostipes*, *Lachnospiraceaeincertaesedis*, and *Phascolarctobacterium*, and decreases in *Ruminococcus*, *Faecalibacterium*, *Bifidobacterium*, *Dialister*, and *Escherichia/Shigella* [30]. At the phylum level, Actinobacteria, Bacteroidetes, Firmicutes, Fusobacteria, and Protobacteria were different between MDD and controls [30]. linear discriminant analysis (LDA) effect size (LEfSe) analysis showed that Prevotellaceae and *Prevotella* showed increased abundance in MDD, whereas Bacteroidaceae, *Bacteroides*, and *uncultured_Mesorhizobium_sp.* showed increased abundance in controls [31].

Functional gut dysbiosis may further contribute to MDD via many different pathways. First, increased abundance of pathobionts may contribute to increased NINONS activities, sympatho-adrenal system activity, metabolic changes, insulin resistance, neurodegenerative processes, and damage to lipoproteins. A second pathway is depletion of anti-inflammatory microbiota and microbiota that keep the epithelial barrier and the gut healthy. These include microbiota that produce alkaline phosphatase, microbiota that break down polysaccharides, LPS, and sulphate, and other beneficial microbiota that generate short-chain fatty acids (SCFAs, such as butyrate) and vitamin antioxidants [32,33,34,35,36,37,38,39,40,41,42,43,44,45,46,47,48,49,50,51,52].

Nevertheless, there are no data on whether, in Thai MDD patients, there are any indicants of gut dysbiosis, as indicated by reduced diversity of gut flora and differential abundance of bacterial taxa. Hence, the present study was performed to delineate whether MDD is characterized by diminished gut bacterial microbiota alpha and beta diversity, changes in relative abundances of gut bacteria at the phylum, genus, and species levels, and differential abundance of bacterial taxa based on LEfSe.

## 2. Materials and Methods

### 2.1. Participants

For this study, thirty-seven normal controls and thirty-two patients with major depressive disorder (MDD) were recruited from the Department of Psychiatry’s outpatient clinic at King Chulalongkorn Memorial Hospital in Bangkok, Thailand. Participants ranged in age from 19 to 58 years old and were of both sexes. They were diagnosed with MDD using DSM-5 criteria. The control group was recruited in the same catchment area as the patients, Bangkok, Thailand, via word of mouth. Control participants having any DSM-5 axis-1 disorder diagnosis or a positive family history of MDD, bipolar disorder (BD), or suicide were excluded from the study. MDD participants having any DSM-5 axis-1 disorder diagnosis other than MDD were excluded from the study, e.g., BD, schizophrenia, schizo-affective disorder, anxiety disorders, obsessive compulsive disorder, post-traumatic stress disorder, autism, substance use disorder (except nicotine dependence), and psycho-organic disorders. Patients and controls were excluded if they had any of the following conditions: (a) (auto)immune diseases, such as chronic obstructive pulmonary disease, cancer, psoriasis, type 1 diabetes, and asthma; (b) inflammatory bowel disease or irritable bowel syndrome; (c) neurodegenerative, neuroinflammatory, or neurological disorders, such as Alzheimer’s disease, multiple sclerosis, stroke, epilepsy, or Parkinson’s disease; (d) inflammatory or allergic reactions three months prior to the study; (e) pregnant or lactating women.

All patients and controls provided written consent before taking part in this study. The study was carried out in compliance with international and Thai ethical standards as well as privacy legislation. The Chulalongkorn University Faculty of Medicine’s Institutional Review Board in Bangkok, Thailand (#446/63) approved the study in accordance with the International Guidelines for the Protection of Human Subjects as required by the Declaration of Helsinki, the Belmont Report, the CIOMS Guidelines, and the International Conference on Harmonization in Good Clinical Practice (ICH-GCP).

### 2.2. Clinical Assessments

Semi-structured interviews were conducted by a trained research psychologist specialized in mood disorder research. To assess the severity of depression symptoms, we employed the Hamilton Depression Rating Scale (HDRS) [52] and the Beck Depression Inventory-II (BDI) [53]. To assess the axis-1 diagnosis, the Mini-International Neuropsychiatric Interview (M.I.N.I.) was utilized [54]. Using DSM-5 criteria, tobacco use disorder (TUD) was diagnosed. Body mass index (BMI) was calculated by dividing body weight (in kilograms) by height squared (in meters).

### 2.3. Assays

#### 2.3.1. Fecal Sample Collection and DNA Extraction

Approximately 20 mg fecal samples were collected in sterile test tubes containing 2 mL of DNA/RNA Shield™ reagent (ZYMO Research, Irvine, CA, USA) and kept at −20 °C until tested. DNA was extracted using ZymoBIOMICS™ DNA Miniprep Kit (ZYMO Research, USA) following the manufacturer’s standard protocol.

#### 2.3.2. 16S rDNA Amplification

The full length of the bacterial 16S rDNA gene (1.5 kb) was amplified by PCR using specific primers, 5′-TTTCTGTTGGTGCTGATATTGCAGRGTTYGATYMTGGCTCAG-3′ and 5′-ACTTGCCTGTCGCTCTATCTTCCGGYTACCTTGTTACGACTT-3′, as described previously [55]. The first round of PCR reaction contained 1 µg of DNA template, 0.2 µM of each primer, 0.2 mM of dNTPs, 1X Phusion™ Plus buffer, 0.4 U of Phusion Plus DNA Polymerase (Thermo Scientific, Waltham, MA, USA), and nuclease-free water in a final volume of 20 µL. The PCR reaction was performed under thermal conditions of 98 °C for 30 s, 25 cycles of amplification (98 °C for 10 s, 60 °C for 25 s, 72 °C for 45 s), and 72 °C for 5 min. After that, the barcodes were attached to the 16S rDNA amplicon by five cycles of amplification (98 °C for 10 s, 60 °C for 25 s, 72 °C for 45 s) based on PCR Barcoding Expansion 1–96 (EXP-PBC096) kit (Oxford Nanopore Technologies, Oxford, UK). The amplicons were purified using QIAquick^®^ PCR Purification Kit (QIAGEN, Hilden, Germany) according to the manufacturer’s protocol.

#### 2.3.3. 16S rDNA Amplicon Sequencing Based on Oxford Nanopore Technology (ONT)

The concentrations of purified amplicons were measured using a Qubit 4 fluorometer with a Qubit dsDNA HS Assay Kit (Thermo Scientific, USA). Then, the amplicons of 63 samples with different barcodes were pooled at equal concentrations (1 μg in the total volume of 48 μL) and purified using 0.5X Agencourt AMPure XP beads (Beckman Coulter, Brea, CA, USA). After that, the purified DNA library was end-repaired and adaptor-ligated using a Ligation Sequencing Kit (SQK-LSK112) (Oxford Nanopore Technologies, UK). Finally, the DNA library (approximately 315 ng) was loaded on the R10.4 flow cell and then sequenced by the MinION Mk1C platform (Oxford Nanopore Technologies, UK).

### 2.4. Data Processing and Statistical Analysis

Guppy basecaller software v6.0.7 [56] (Oxford Nanopore Technologies, UK) was used for base-calling with a super-accuracy model to generate pass reads (FASTQ format) with a minimum acceptable quality score (Q > 10). The quality of reads was examined by MinIONQC [57]. Then, FASTQ sequences were demultiplexed and adaptor-trimmed using Porechop v0.2.4 [58]. The filtered reads were then clustered, polished, and taxonomically classified by NanoCLUST [59] based on the full-length 16S rRNA gene sequences from the Ribosomal Database Project (RDP) database [60]. Rarefaction analysis was performed to estimate the minimum number of reads (2000 reads per sample) considered adequate for further analysis. The abundance taxonomic assignment data were converted into QIIME data format to illustrate bacterial richness and evenness based on their taxa abundances and alpha diversity (Chao1 and Shannon indexes). Then, the beta diversity was analyzed with the Bray–Curtis cluster analysis index using a plug-in implemented for QIIME2 software v2021.2 [61]. The normalized data were visualized by MicrobiomeAnalyst [62]. The differential abundance was analyzed based on a LEfSe analysis with *p* < 0.05 and linear discriminant analysis (LDA) score > 2 [63] using the Galaxy server [64]. Isometric log-ratio (ILR) Box–Cox transformation was applied to the microbiota abundance data at the phylum, genus, and species levels (ILR abundance). Multiple regression analysis was performed to examine the effects of the actual MDD diagnosis on the relevant (LEfSe significant) ILR-transformed abundance data, while allowing for the effects of age, sex, and body mass index. The significance level was set at 0.05, two-tailed. We used IBM SPSS, Windows version 28, to analyze the data.

## 3. Results

### 3.1. Socio-Demographic and Clinical Data

Table 1 displays the socio-demographic data of the participants divided into controls and MDD patients. There were no differences in age, sex ratio, education, BMI, MetS, or employment among both groups. There was a trend to a lower rate of married people, and increased TUD rate, in MDD patients as compared with controls. The HDRS and BDI scores were significantly higher in MDD patients than in controls. Some of the MDD patients were treated with psychotropic drugs, namely, sertraline (*n* = 10), fluoxetine (*n* = 6), escitalopram (*n* = 6), trazodone (*n* = 4), benzodiazepines (*n* = 11), antipsychotic agents (*n* = 5), and mood stabilizers (*n* = 1). Statistical analyses were used to examine if the drug state of the patients might affect the microbiome features. However, even without false-discovery rate *p* correction, no significant effects of these psychotropic drugs could be found on the relevant ILT-transformed microbiota data (at the phylum, genus, and species level, using regression analyses).

### 3.2. Gut Bacterial Microbiota Diversity

The full-length bacterial 16S rDNA was sequenced based on high-throughput long-read nanopore sequencing, providing 827,392 raw reads in total, with an average of 13,133 reads per sample. After quality filtering, the retained reads were classified into operational taxonomic units (OTUs). Alpha diversities (Chao1 and Shannon indexes) were used to demonstrate the richness and evenness of bacterial comparisons between the control and MDD groups based on their relative abundances. There were no significant differences using the Mann–Whitney U test between both study groups, as shown in Figure 1A,B. Based on a Bray–Curtis dissimilarity index to compare the bacterial communities between the control and MDD groups (Figure 1C), the result showed no significant differentiation with identity at a 95% confidence interval.

### 3.3. Gut Bacterial Profile and Composition

The gut bacterial profiles were characterized by the microbial composition of samples in the control and MDD groups. Figure 2 shows that the most abundant phylum was Firmicutes, followed by Bacteroidetes and Proteobacteria in both groups. The top three most abundant bacteria at the genus level were *Faecalibacterium*, *Prevotella*, and *Bacteroides* in the controls (68.9%) and the MDD group (54.7%). Additionally, relative abundance in species level between groups could not be clearly distinguished. However, *Faecalibacterium prausnitzii*, *Prevotella copri*, *Oscillibacter valericigenes*, *Bacteroides vulgatus*, and *Prevotella stercorea* were the most prevalent gut bacteria in the controls (61.3%) and the MDD group (41.9%).

### 3.4. Differential Abundance of Gut Bacteria between the Control and MDD Groups

LEfSe analysis was performed to classify the significant differences in bacteria between the control and MDD groups based on linear discriminant analysis (LDA) scores (>2), as shown in the bar graph (Figure 3A) and cladogram (Figure 3B). The Rhodospirillaceae family, *Hungatella* genus, and bacterial species, including *Clostridium bolteae*, *Hungatella hathewayi*, and *Clostridium propionicum*, were significantly enriched in the MDD group. *Desulfovibrio piger*, *Ruminococcus callidus*, *Coprococcus comes*, *Gemmiger formicilis*, and *Phascolarctobacterium succinatutens* were enriched in the control group. We found that the diagnosis of MDD (versus controls) explained 9.8% of the variance in the ILR-transformed *R. callidus* abundance values. Sex, age, body mass index, and the drug state of the patients (use of antidepressants, benzodiazepines, and mood stabilizers) were not significant in this regression analysis. There were no significant effects of sex, age, or the drug state of the patients on any of the relevant (LEfSe-significant) ILR-transformed abundance data.

## 4. Discussion

### 4.1. Key Findings of Our Study

The major findings of this study are that: (a) there are no significant differences in α- or β-diversity between MDD and controls; (b) there is no clear difference in relative abundance of species between MDD and controls; and (c) LEfSe analysis revealed that the Rhodospirillaceae family (Gram-negative, rod-shaped to spirillum-formed, purple non-sulfur bacteria which comprise 34 genera; [36,37]), *Hungatella* genus (anaerobic, Gram-positive bacterial genus from the family of *Clostridiaceae*; [38]), and the species *Clostridium bolteae* (anaerobic, Gram-positive, rod-shaped, spore-forming bacteria, [39]), *Hungatella hathewayi* (anaerobic, Gram-positive bacterium, [40]) and *Clostridium propionicum* (anaerobic, Gram-positive, rod-shaped bacteria, [41]) were significantly enriched in the MDD group. In contrast, *Desulfovibrio piger (aerotolerant*, *Gram-negative bacteria*, [42]), *Ruminococcus callidus* (anaerobic, Gram-positive bacteria; [47]), *Coprococcus comes* (anaerobic Gram-positive, cocci, [43]), *Gemmiger formicilis* (anaerobic weakly Gram-positive bacteria; [65]), *Phascolarctobacterium succinatutens* (strictly anaerobic, Gram-negative bacteria; [45]), *Lutispora* (anaerobic, Gram-stain-negative and Gram-positive cell-wand structure; [66]), and Gracillibacteraceae (belonging to the Firmicutes phylum) are enriched in the control group.

### 4.2. α-Diversity and MDD

Our negative findings regarding α-diversity corroborate some prior research papers. Four reviews on α-diversity in MDD or mixed groups of the major psychiatric diseases found no convincing evidence of changes in α-diversity of bacteria in MDD [67,68,69,70]. According to the most recent systematic review, there was a greater number of studies that found no differences (*n* = 8) or inconsistent results (*n* = 7) than those that revealed reduced α-diversity (*n* = 5). In MDD, three research papers found no differences, four studies observed inconsistencies, and two studies found decreased α-diversity [67]. According to a recent systematic review [71], 14 out of 21 studies identified no significant variations in α-diversity between MDD and controls. McGuinness et al. [70] stated that there is no convincing evidence that patients with mental illnesses show less α-diversity. Ritchie et al. [72] and Thapa et al. [73] reported that there are no significant differences in the α-diversity of bacteria between patients with MDD and controls, whereas other studies reported reduced [74] or increased α-diversity [75] in MDD. Ye et al. [76] determined, using the Chao1 and Shannon indices, that α-diversity was greater in MDD than in controls. In their investigation, Zhang et al. [31] discovered, using the Simpson and Pielou’s index, that the α-diversity was reduced in MDD patients without adverse childhood experiences, but no differences were found between controls and MDD patients with adverse childhood experiences. Overall, the results imply that microbiome α-diversity is not significantly altered in MDD.

### 4.3. β-Diversity and MDD

Our negative findings regarding β-diversity corroborate previous studies. The recent systematic review by Borkent et al. [67] showed that three research papers did not identify differences in β-diversity between MDD and controls, while four investigations reported decreased β-diversity [67]. Inconsistent β-diversity findings were also presented in the review by Simpson et al. [68]. In their systematic review, Alli et al. [71] found that 12 out of 18 studies demonstrated that β-diversity was significantly different between MDD and controls. Moreover, McGuinness et al. [70] delineated that studies on β-diversity in mental diseases, including MDD, are reasonably consistent. Some more recent studies have revealed contradictory findings regarding β-diversity in MDD. Thus, Ritchie et al. [72] and Thapa et al. [73] were unable to discover significant changes in β-diversity in MDD and adolescent depression versus controls, respectively. Liu et al. [74] observed a reduction in both α- and β-diversity of the gut microbiota composition. Sun et al. [75] discovered variations in β-diversity between MDD and controls and found that, at *p* = 0.03, the between-group differences were greater than the within-group differences. Zhang et al. [31], in their β-diversity study, found significant differences between controls and MDD. Overall, our findings add to the negative research findings on β-diversity in MDD, which account for roughly one-third of all studies.

### 4.4. Abundance of Gut Microbiome Taxa in MDD

Borkent et al. [67] reported that all research on MDD and other severe psychiatric diseases, such as schizophrenia and BD, reported statistically significant variations in the relative abundance of bacterial taxa. Nonetheless, it appears that these distinctions are sometimes extremely inconsistent, whilst the most consistent results were increased abundance of *Lactobacillus*, *Streptococcus*, and *Eggerthella* in controls, and decreased abundance of anti-inflammatory butyrate-producing *Faecalibacterium* in the combined group of mental disorders. Our assessment of the signature of gut microbiota utilizing LEfSe analysis to find the classes of organisms that explain differences across diagnostic groups did not support the conclusions of Borkent et al. [67]. Our review shows that the microbiome LEfSe profile of MDD has only a few parallels with previous LEfSe investigations in MDD.

For instance, Zhang et al. [37] revealed that 27 microorganisms were linked to MDD or controls. These authors found that Ruminococcaceae family, *Gemmiger*, and *Phascolarctobacterium* were enriched in controls, which partly corresponds with our results showing that *Gemmiger*, *Gemmiger f.*, and *Phascolarctobacterium s*. were enriched in controls. In another study, multi-omics studies of the gut microbiome in MDD and LEfSe analysis identified numerous microbiome characteristics in MDD. However, none of these correlated with our findings [77]. In addition, whereas we observed an increase in the prevalence of *Ruminococcus* and *Clostridium* in the control group, Zhao et al. [77] observed an increase in the prevalence of some *Ruminococcus* and *Clostridium* taxa in MDD. The LEfSe analysis in the study by Liu et al. [74] revealed that Deinococcaceae were enriched in MDD, whereas Clostridiaceae, Bacteroidaceae, Turicibacteraceae, and Barnesiellaceae were enriched in controls. At the genus level, *Deinococcus* and *Odoribacter* were associated with MDD, and *Clostridium*, *Bacteroides*, *Alistipes*, *Turicibacter*, *Roseburia*, and *Enterobacter* with controls. Thus, the sole resemblance between Liu’s study and ours is the higher relative abundance of *Clostridium* in the control group. We, and Jiang et al. [78], both observed increased abundance in *Ruminococcus* in controls versus MDD patients. LEfSe analysis performed on patients with depression and anxiety in screening for gastro-intestinal cancer confirmed that *Gemmiger* and *Ruminococcus* were overabundant in controls, but none of the other taxa were discriminant features [79]. In a study examining the microbiome in a major depressive episode associated with BD, LEfSe analysis revealed differentially abundant features, including increased abundance of the phylum Actinobacteria and the class Coriobacteria, whereas *Faecalibacterium* genus and Ruminococcaceae family were more abundant in controls [80]. These latter findings corroborate our results that the *Ruminococcaceae* family is overexpressed in controls. Another study discovered fecal microbiome characteristics that differentiate childhood depression [81]. Although this LEfSe analysis discovered numerous discriminative taxa at an LDA threshold of > 3, there were no parallels between our findings and those of Ling et al. Moreover, whilst we discovered that *Gemmiger* and *Desulfovibrio* were abundant in controls, Ling et al. [81] discovered the opposite. There are no similarities between our results (obtained in younger MDD patients and controls) and those of patients with late-life depression [82], despite the fact that the latter LEfSe analysis found many significant abundances of bacterial taxa at the genus level in controls as opposed to patients with late-life depression.

Overall, a comparison of the LEfSe analysis undertaken in MDD reveals few parallels between the various investigations. As microbiome compositum is heavily impacted by diet [83], it is likely that microbiome features of a human disorder are culture-specific and may not always correspond with the profiles established in other countries. As a consequence, the microbiome profile shown here may be more specific to Thai MDD patients.

### 4.5. Is There Compositional Gut Dysbiosis in MDD?

Certainly, it is plausible to hypothesize that the dysbiotic microbiome features identified by our LEfSe analysis contribute to the pathophysiology of MDD by: (a) increasing the predominance of some potential pathobionts, such as Rhodospirillaceae (inhibiting calpain protease, which plays a role in neurodegenerative processes), *C. propionicum* (may induce the sympatho-adrenal system with increased norepinephrine production and consequent insulin resistance), and *H. hathewayi* (possible pathogen that is associated with colorectal cancer); and (b) lowering the abundance of protective microbiota including *Clostridium* (promotes SCFA production), *Coprococcus* (involved in the maintenance of a healthy gut and epithelial barrier function), *Desulfovibrio* (reduces sulphate production, which may be related to many physio-somatic symptoms when upregulated), *Phascolarctobacterium succinatutens* (production of SCFA including propionate), Gracillibacteraceae (association with attention deficit disorder), and *Gemmiger* and *G. formicillis* (production of SCFA; depletion is associated with inflammatory bowel disease) [36,37,38,39,40,41,42,43,44,45,46,65].

These data may imply compositional gut dysbiosis in MDD, which is defined as an increase in potentially harmful enterobacteria and a decrease in gut beneficial microbiota. However, it remains unknown if these changes in pathobionts and beneficial microbiota reflect functional gut dysbiosis and whether dysbiosis is a cause or effect of the NINONS responses in MDD. If these results are indicative of functional dysbiosis, they are likely unique to Thai patients with MDD. In fact, distinct microbiome profiles in distinct cultures may result in comparable functional effects, either pathological or beneficial. For example, there are even differences in microbiota composition among Italian participants from different regions in Italy [84]. These authors recommend a region-specific strategy for microbiota engineering [84]. It must be stressed that the current study recruited participants living in Bangkok, Thailand, and not in other regions of Thailand.

### 4.6. Is Depletion of Ruminococcus Important to MDD?

The most consistent finding in the reviewed literature is perhaps the higher relative abundance of *Ruminococcus* in controls versus MDD patients [37,78,79,80], whereas one study reported contradictory results [77]. Additionally, utilizing bidirectional Mendelian randomization, Chen et al. [85] discovered that *Ruminococcus* protects against MDD. Other studies in rodent models indicate that a decrease in *Ruminococcus* or some *Ruminoccocus* species is linked to depression or antidepressant effects [86,87,88]. Interestingly, the abundance of *Ruminococcus* is significantly associated with 5-HT levels in a rodent model of postpartum depression [89]. Different *Ruminococcus* species are associated with NINONS-associated human disorders that exhibit a strong comorbidity with MDD [90], including chronic fatigue syndrome, inflammatory bowel disease, type 1 and 2 diabetes mellitus, atherosclerosis and cardiovascular disease, and nonalcoholic fatty liver disease (NAFLD) [46]. In Parkinson’s `disease, those with cognitive impairment had a lower abundance of the genus *Ruminococcus* than those without cognitive abnormalities [91]. *Ruminococcus* genus and *R. callidus* are much less prevalent in the feces of patients with inflammatory bowel disease [48].

The *Ruminoccocus* genus is a key symbiont of the core gut microbiota and plays an important role in gut health via degradation and conversion of polysaccharides (e.g., cellulose and starch) into nutrients which provide cross-feeding to other microbiota [47,48]. In addition, species of this genus display cellulolytic activity and may degrade LPS. Furthermore, different *Ruminococcus* species produce alkaline phosphatase [34], which improves gut barrier functions and detoxifies LPS, thereby preventing activation of NINONS pathways [49,50]. One could, therefore, hypothesize that depletion of this genus may have contributed to lowered LPS breakdown and thus increased translocation of LPS in MDD particularly when deficits in tight junctions are present [13,14,15,16]. *R. callidus* is consistently present in the healthy human gut and probably plays a role in the maintenance of a healthy gut environment [48]. As such, depletion of *Ruminococcus* could constitute a more universal feature of human MDD.

### 4.7. Limitations

It would be fascinating to assess the functional repercussions of microbiome changes in MDD using shotgun metagenomic sequencing in conjunction with pathway enrichment analysis. Importantly, future research should employ Kegg pathway analysis to further delineate gut LPS biosynthesis in MDD as a risk factor contributing to the increased translocation of Gram-negative bacteria. Moreover, future prospective studies should explore the microbiome before and after the development of the first depressive episode in young adults at risk for developing depression (e.g., those with greater childhood adverse experiences and increased neuroticism scores). Another issue is that sex-related changes in the microbiome could, in theory, be associated with the higher prevalence of MDD in women [92]. Nevertheless, our study did not provide any evidence of sex-related differences in the microbiota abundance data, which were significant discriminators between MDD and controls. In addition, the sex ratio was not different between both study groups. Moreover, other pathways, including immune–serotonin interactions, are factors that determine the greater prevalence of depression in women [92]. Future research should examine the effects of treatment with antidepressants on the microbiome in association with the clinical response.

## 5. Conclusions

The levels of α- and β-diversity were similar between those with MDD and controls. The LEfSe analysis revealed an increased abundance of some pathobionts and a decrease in some beneficial gut commensals. Although these results may theoretically suggest compositional gut dysbiosis in MDD, previous studies have been unable to delineate the same microbiome features and some have even shown contradictory results. If our results would be replicated in other Thai samples, the LEfSe profile established here is likely to be unique to Thai patients with MDD. The depletion of *Ruminococcus* has been observed in some human studies (although some studies reported contradictory results), animal models, and disorders that are comorbid with MDD and, therefore, an important takeaway from this study is that low levels of *Ruminococcus* may perhaps have a role in MDD. *Ruminococcus* and the species *R. callidus* serve a useful role in the maintenance of a healthy gut environment. For example, they degrade polysaccharides into nutrients and may degrade LPS, functions that could theoretically increase preexisting deficiencies in the tight and adherens junctions of the paracellular pathway. In our opinion, the latter aberrations are secondary to the activation of NINONS-pathways in MDD, but depletion of beneficial gut-commensal genera and species may contribute to leaky gut, and LPS or bacterial translocation.

## Figures and Tables

**Figure 1 cells-12-01240-f001:**
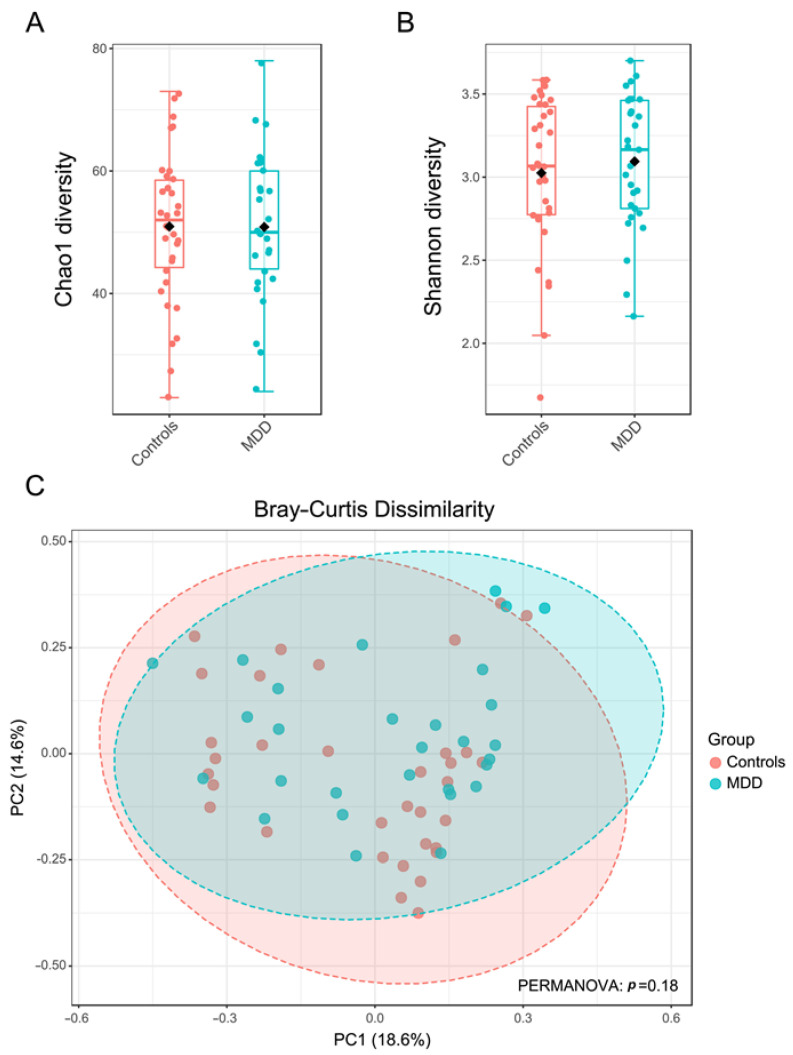
Gut bacterial microbiota diversity in the control (pink) and MDD (blue) groups. The alpha diversity comparison calculated by Chao1 (**A**) and Shannon (**B**) indexes are shown as box plots, with the error bars representing the standard deviation and calculated statistically significant difference by Mann–Whitney U test. The beta diversity was presented by principal coordinate analysis (PCoA) plots based on Bray–Curtis distance (**C**). Permutational ANOVA (PERMANOVA) was applied to calculate statistical differences in beta diversity.

**Figure 2 cells-12-01240-f002:**
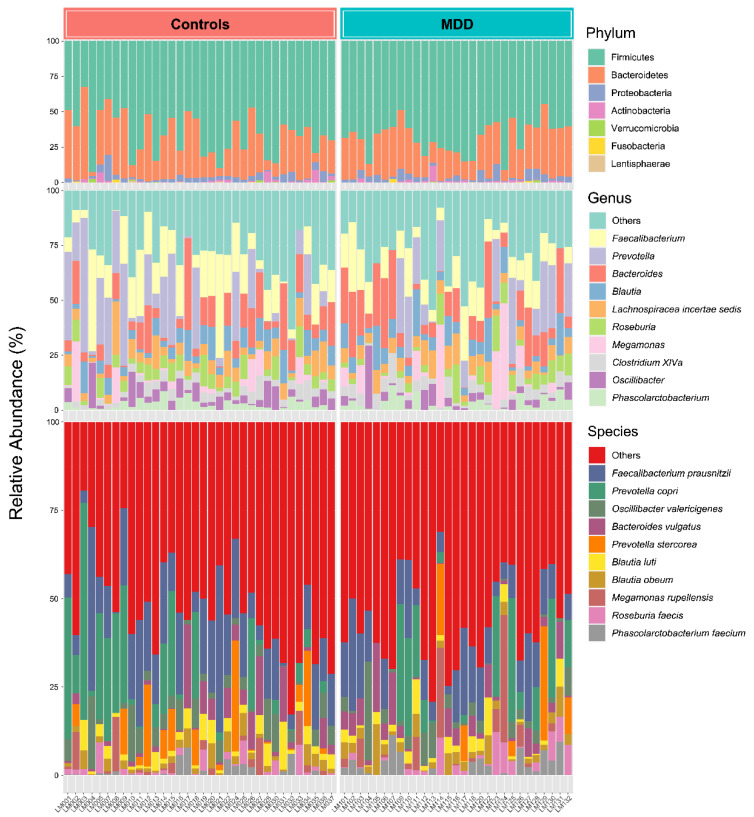
The stacked plot shows relative abundances of gut bacteria at the phylum, genus, and species levels between the control and MDD groups. The data were normalized by a total sum scaling (TSS) method. The colored bars represent various bacterial taxa identified in the gut bacterial profiles.

**Figure 3 cells-12-01240-f003:**
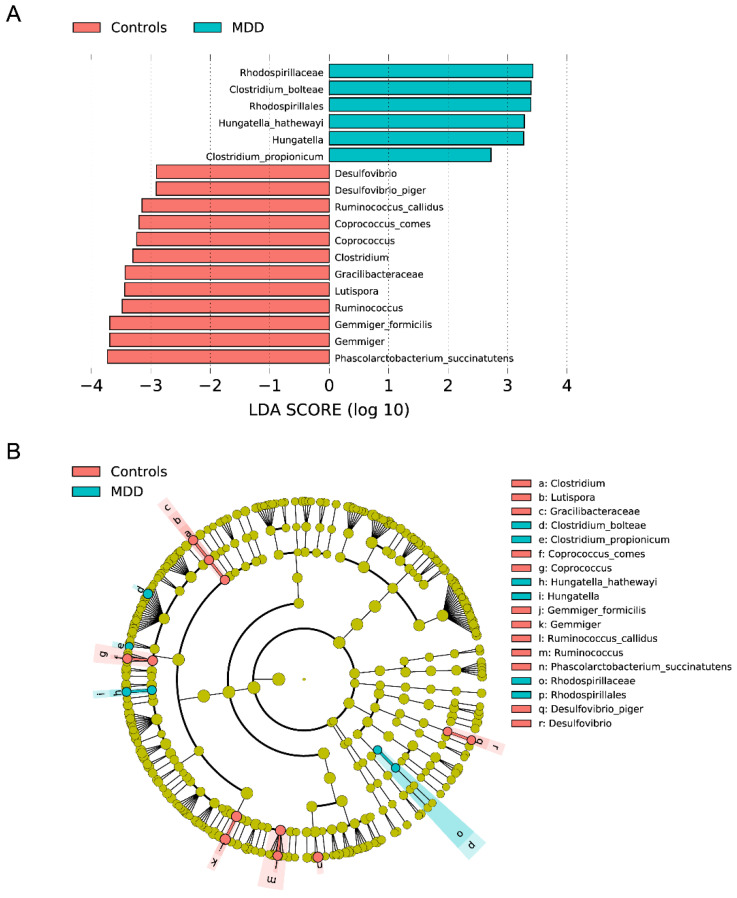
The differential abundance of bacterial taxa between the control and MDD groups based on linear discriminant analysis (LDA) effect size (LEfSe) analysis. LDA scores significantly enriched in the differentially taxonomic levels among groups are represented as a bar graph (**A**) and cladogram (**B**).

**Table 1 cells-12-01240-t001:** Socio-demographic and clinical data of patients with major depressive disorder (MDD) and healthy controls (HC) included in the present study.

Variables	HC N = 37	MDD N = 32	F/χ^2^	df	*p*
Age (years)	28.4 (6.9)	25.9 (9.1)	1.61	1/67	0.209
Female/Male ratio	31/6	26/6	0.08	1	1.00
Education (years)	16.0 (2.1)	16.2 (3.2)	0.15	1/67	0.703
Employment (No/Yes)	0/37	1/31	FEPT	-	0.464
Single/Married	29/8	31/1	FEPT	-	0.031
TUD (No/Yes)	35/2	24/8	5.31	1	0.037
BMI	22.6 (4.9)	24.7 (5.8)	2.91	1/67	0.093
MetS	32/5	29/3	0.29	1	0.716
HDRS	1.8 (1.8)	15.7 (5.2)	232.37	1/67	<0.001
BDI	5.8 (7.3)	23.6 (11.6)	59.96	1/67	<0.001

Results are shown as mean (SD) or as a ratio; F: results of analyses of variance; χ^2^: results of analyses of contingency tables. TUD: tobacco use disorder; BMI: body mass index; MetS: metabolic syndrome; HDRS: Hamilton Depression Rating Scale; BDI: Beck Depression Inventory.

## Data Availability

The dataset generated during and/or analyzed during the current study will be available from MM upon reasonable request and once the authors have fully exploited the dataset.

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
