# Peer review of "Exploration of the Gut Microbiome in Thai Patients with Major Depressive Disorder Shows a Specific Bacterial Profile with Depletion of the Ruminococcus Genus as a Putative Biomarker"

_cells, 2023, doi:10.3390/cells12091240_

Round 1

Reviewer 1 Report (New Reviewer)

Depressive disorders are associated with gut dysbiosis that disrupts the gut-brain axis. However, the link between gut microbiota and changes in brain structures observed in depressed patients remains still elusive. Research in this subject should aim to establish correlations between neuropsychiatric parameters, changes in the structure of brain areas, the composition of the microbiota in the digestive system and broadly defined immunological parameters defining the balance between inflammatory and anti-inflammatory factors. Thus, in addition to the conventional approach of characterizing the pathophysiology of depression in the context of neurotransmitter disorders, stress hormones, neurotrophic factors and pro-inflammatory cytokines, an analysis of the composition of the intestinal microbiota and secreted agents, an important element influencing the functioning of the immune system and maintaining body homeostasis through neuro-endocrine-immuno-brain interactions is proposed. Comprehensive research is possible using experimental models on mice. An important element of research should also be the search for a relationship between the composition of the microbiota and the effectiveness of pharmacological therapy.

In this aspect, the research results presented by the Authors of the paper are of great importance. Additional value is provided by the research results in connection with ethnic/geographical diversity, which, as might be expected, is related to regional diet and broadly understood lifestyle.

In the 64th line of the manuscript, the order of the letters in the abbreviation LPS  has been changed.

The Authors chose very well the methods of research and statistical analysis. The critical discussion of the obtained research results should be highly appreciated.

Author Response

An important element of research should also be the search for a relationship between the composition of the microbiota and the effectiveness of pharmacological therapy.

@@ANSWER: addressed in the text as:

Future research should examine the effects of treatment with antidepressants on the microbiome in association with the clinical response.

In the 64th line of the manuscript, the order of the letters in the abbreviation LPS  has been changed.

@@ANSWER: corrected.

Reviewer 2 Report (New Reviewer)

The authors analysed alterations in faecal microbiota in MDD patients of Tai population, with regard to other studies also examining microbiota changes related to MDD. The article is well written, and results are well presented and discussed.

However, the speculation that the decrease of Ruminoccocus could contribute to LPS load in MDD is not well supported by the literature and should be toned down, especially in the conclusion. Further, the authors state that Xiao Q. et al. (ref. 87.) showed that chronic stress decreased Ruminoccocus while treatment with crocin-I increased it in parallel with its antidepressant properties (lines 415-420). However, their results are exactly the opposite. Moreover, there are other studies in rodent models showing that decrease of at least some of Ruminoccocus species is linked to antidepressant effects (e.g. Burokas et al, 10.1016/j.biopsych.2016.12.031; Lukic et al, 10.1038/s41398-019-0466-x). Therefore, you can comment these inconsistencies as well.

Minor comments:

1.       Line 64 - change „PLS“ to „LPS“

2.       The reference 38. „Zhang, W., Lu, L., Lai, Q., Zhu, B., Li, Z., Xu, Y., Shao, Z., Herrup, K., Moore, B.S., Ross, A.C., Qian, P.Y. Family-wide Structural Characterization and Genomic Comparisons Decode the Diversity-oriented Biosynthesis of Thalassospiramides by Marine Proteobacteria. J Biol Chem.“ is cited wrongly. Please correct this.

3.       Lines 347–348 – Please, clarify the sentence: „On the other hand, neither study discovered any similarities in the taxa related to MDD“ since in the following text you mention studies with some similarities with your results.

4.       Lines 382–383 – Please, clarify the sentence: “Therefore, the micriobiome profile shown here may be unique to Thai individuals with MDD, except maybe for the Ruminococcus genus which has been detected in other countries”, and specify that Ruminococcus genus was increased in the control group.

5.       Line 440 - change „Reminococcus” to “Ruminococcus”.

Author Response

However, the speculation that the decrease of Ruminoccocus could contribute to LPS load in MDD is not well supported by the literature and should be toned down, especially in the conclusion.

@@ANSWER: addressed as, see last sentence of the paper:

Ruminococcus and the species R. callidus serve a useful role in the maintenance of a healthy gut environment, for example, they degrade polysaccharides into nutrients and may degrade LPS, functions that could theoretically increase preexisting deficiencies in the tight and adherens junctions of the paracellular pathway. In our opinion, the latter aberrations are secondary to the activation of NINONS-pathways in MDD, but depletion of beneficial gut commensal genera and species may contribute to increased LPS or bacterial translocation.

Further, the authors state that Xiao Q. et al. (ref. 87.) showed that chronic stress decreased Ruminoccocus while treatment with crocin-I increased it in parallel with its antidepressant properties (lines 415-420). However, their results are exactly the opposite. Moreover, there are other studies in rodent models showing that decrease of at least some of Ruminoccocus species is linked to antidepressant effects (e.g. Burokas et al, 10.1016/j.biopsych.2016.12.031; Lukic et al, 10.1038/s41398-019-0466-x). Therefore, you can comment these inconsistencies as well.

@@ANSWER: I reread the Xiao paper many times (Abstract and Results and discussion) and I must admit I do not understand what the authors want to convey. So, I deleted this sentence but now write:

Other studies in rodent models indicate that the decrease of Ruminococcus or some Ruminoccocus species are linked to depression or antidepressant effects [87,88,89].

Minor comments:

  1. Line 64 - change „PLS“ to „LPS“

@@ANSWER : Corrected.

  1. The reference 38. „Zhang, W., Lu, L., Lai, Q., Zhu, B., Li, Z., Xu, Y., Shao, Z., Herrup, K., Moore, B.S., Ross, A.C., Qian, P.Y. Family-wide Structural Characterization and Genomic Comparisons Decode the Diversity-oriented Biosynthesis of Thalassospiramides by Marine Proteobacteria. J Biol Chem.“ is cited wrongly. Please correct this.

@@ANSWER: I checked, but it is OK. This paper reviews Rhodospirillaceae.

  1. Lines 347–348 – Please, clarify the sentence: „On the other hand, neither study discovered any similarities in the taxa related to MDD“ since in the following text you mention studies with some similarities with your results.

@@ANSWER: I have deleted: “On the other hand, neither study discovered any similarities in the taxa related to MDD“.

  1. Lines 382–383 – Please, clarify the sentence: “Therefore, the micriobiome profile shown here may be unique to Thai individuals with MDD, except maybe for the Ruminococcus genus which has been detected in other countries”, and specify that Ruminococcusgenus was increased in the control group.

@@ANSWER: changed into: The micriobiome profile shown here may be more specific to Thai MDD patients.

  1. Line 440 - change „Reminococcus” to “Ruminococcus”.

@@ANSWER corrected

Reviewer 3 Report (New Reviewer)

Dear Authors,

The study is well-designed and presented with all the necessary information and adequately selected controls. I have only minor suggestions for improvement.

As the microbiome studies employing full-length 16S rRNA ONT sequencing are relatively new, I believe some more details could be included in the Materials and methods for future reference. For example:

- how many samples were multiplexed on a single 10.4 cell/ run?

- what concentration/molarity was considered for pooling of the barcoded amplicons

- what was the approximate DNA amount loaded on the flow cells?

- what was the minimum number of reads per sample considered adequate for analysis?

Best regards

Author Response

The study is well-designed and presented with all the necessary information and adequately selected controls. I have only minor suggestions for improvement. As the microbiome studies employing full-length 16S rRNA ONT sequencing are relatively new, I believe some more details could be included in the Materials and methods for future reference. For example:

  1. How many samples were multiplexed on a single 10.4 cell/ run?

@@ANSWER: addressed in the text as:

In this study, 63 samples were multiplexed on a single flow cell version 10.4.

  1. What concentration/molarity was considered for pooling of the barcoded amplicons?

@@ANSWER: addressed in the text as:

The amount of 1 mg in the total volume of 48 mL was considered for pooling the barcoded amplicons.

  1. What was the approximate DNA amount loaded on the flow cells?

@@ANSWER: addressed in the text as:

Approximately 315 ng of DNA library was loaded on the flow cells.

  1. What was the minimum number of reads per sample considered adequate for analysis?

@@ANSWER: addressed in the text as:

Rarefaction analysis was performed to estimate the minimum number of reads (2,000 reads per sample) considered adequate for further analysis.

Round 2

Reviewer 2 Report (New Reviewer)

Dear authors, thank you for your responses and corrections.

Only, the reference Zhang et al (38) is cited wrongly in the context of MDD patients, lines 347-349.

Reviewer 3 Report (New Reviewer)

Thank you for addressing my suggestions. I do not have further comments. 

Best regards

This manuscript is a resubmission of an earlier submission. The following is a list of the peer review reports and author responses from that submission.

Round 1

Reviewer 1 Report

In this manuscript, the authors conducted a clinical experiment comparing the differences in gut microbiota between Thai healthy people and MDD patients. This article reported that the levels of α- and β-diversity were similar between those with MDD and controls. The LEfSe analysis revealed an increased abundance of some pathobionts and a decrease in some beneficial gut commensals. The most important take away from this study is that low levels of Ruminococcus may have a role in MDD. Ruminococcus and the species R. callidus serve a useful role in the maintenance of a healthy gut environment: they degrade polysaccharides into nutrients and degrade LPS, functions that could theoretically increase LPS translocation in the presence of preexisting deficiencies in the tight and adherens junctions of the paracellular pathway.

The following issues need to be explained or corrected:

1. Remarkably, MDD patients are predominantly female. So in MDD patients, are there any differences in the microbiomes of men and women? What causes the predominance of women in MDD?

2. The author believes that regional differences are the reason why some microbial results are different from the previous literature. Please further elaborate on this point of view.

3. It is recommended that the author supplement the correlation analysis between Ruminococcus and patients' MDD scores.

Overall, this is an interesting study. I think the manuscript can be accepted after moderate revision.

Reviewer 2 Report

Dear authors

This article is well written with appropriate analysis techniques. Nevertheless the research methodology is a source of multiple important biais. The most important restriction is the important variability in the age, treatment of the patients and the duration of the disease. 

Untreated patient should be recruited and compared with healthy aged-matched controls living in the same family / house. Microbiotas should be compared before and after intervention (microbiota-based of pharmacology-based) in line with clinical outcome.